# Linking Experimental and Numerical Wave Modelling

**Sanne van Essen \*, Jule Scharnke, Tim Bunnik, Bülent Düz, Henry Bandringa, Rink Hallmann and Joop Helder**

Maritime Research Institute Netherlands (MARIN), Haagsteeg 2, 6708 PM Wageningen, The Netherlands; j.scharnke@marin.nl (J.S.); t.bunnik@marin.nl (T.B.); b.duz@marin.nl (B.D.); h.bandringa@marin.nl (H.B.); r.hallmann@marin.nl (R.H.); j.helder@marin.nl (J.H.)

\* Correspondence: s.v.essen@marin.nl

**Abstract:** Experimental or numerical analysis of the response of ships and other floating structures starts with correct environmental modelling. The capabilities of numerical tools are rapidly expanding, but presently the evaluation of extreme events in waves (such as slamming, green water, air-gap exceedance) still requires a combination of experiments and different levels of numerical tools. The present paper describes recent efforts within the Maritime Research Institute Netherlands (MARIN) to improve experimental and numerical wave modelling and especially their combination. The ultimate objective is to be able to reproduce any wave condition from a basin or from sea in numerical tools and vice versa, including a sound treatment of basin effects, numerical effects and statistical variability. The aspects that are of importance in both types of wave modelling are first introduced, after which a number of examples of recent projects is discussed. It can be concluded that important steps were made towards linking experimental and numerical wave modelling, but there are some challenges common to all wave reproductions. Some future planned studies focussing on how to deal with them are discussed as well.

**Keywords:** wave modelling; experiments; calculations; validation; basin effects; CFD; propagation; reproduction; events; kinematics

---

## 1. Introduction

Environmental conditions form the starting point for analyses of the behaviour of ships and offshore structures. A correct generation (both numerically and experimentally) of the most realistic representation of the ocean wave field is therefore of greatest importance. Such analysis should start with a definition of 'reality' based on full-scale wave data, including critical assessment of the instrumentation and analysis used to obtain the data (such as the studies of the accuracy of wave measurement equipment by [1–5] or the study of the applicability of directional wave analysis methods on shallow water by [6]). This should be followed by an analysis of the inherent limitations of the basin wave fields and numerical wave generation, considering the most relevant aspects of the wave field for the studied vessel or structure (as recommended by e.g., [7,8] and shown for a case study by [9,10]). All relevant aspects of the wave field have to be modelled at sufficient accuracy. A similar reasoning applies to current and wind conditions and their interactions with the wave field, but the present publication focuses on wave generation only.

Numerical analysis of marine structures has made significant progress in recent years. However, for very nonlinear phenomena such as extreme wave impacts, model tests are still required to derive design loads. Computational Fluid Dynamics (CFD) has increasing capabilities even for violent nonlinear wave impact events (see example in Figure 1), but its computational efficiency is presently not sufficient to evaluate multiple three-hour wave realisations or even full scatter diagrams. The reproduction of (basin) waves in CFD has led to new opportunities [11], such as the possibility to analyse different design configurations based on a base case model test result. On the other hand, in the future, it could be efficient to carry out a large part of the analysis numerically in fast 'screening tools' to identify wave events that are potentially critical for wave impacts and then generate these isolated events in more advanced tools such as CFD or in a model test basin. Such an approach was evaluated for structures at zero speed [12,13]. Smart combinations of different levels of numerical tools and experiments seem to be the most efficient way to handle design loads in extreme wave events. Thus, for future design, it is essential to be able to reproduce basin waves numerically and vice versa.

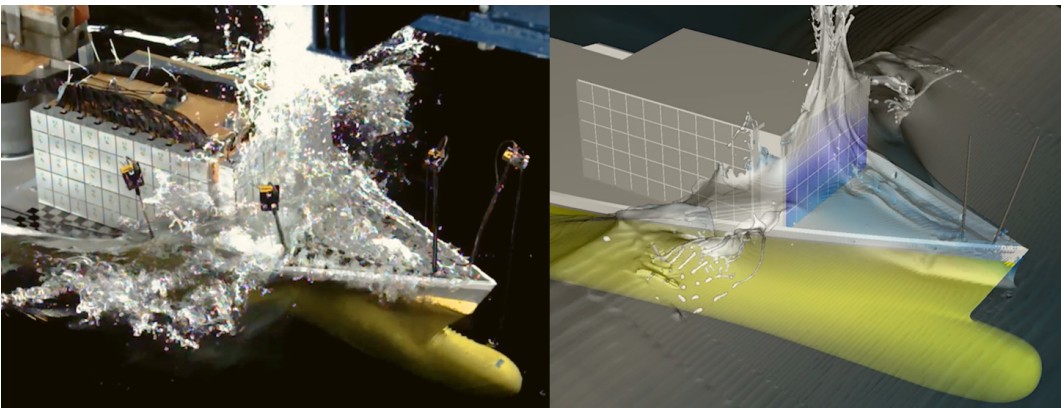

**Figure 1.** Numerical reproduction of green water experiment (more details in Section 4.1).

The present publication provides an overview of recent developments in advanced basin and numerical wave modelling, and the foreseen path towards linking them. This publication first provides an overview of the aspects that are important in basin and numerical wave modelling (Sections 2 and 3) and then discusses a number of recent examples in Section 4. These include numerical reproduction of a single extreme green water event at forward speed in Section 4.1, reproduction of longer wave realisations measured at zero and forward speed in Section 4.2, reproduction of the wave kinematics in breaking waves in Section 4.3 and a discussion on the scale effects in (breaking) wave crests in Section 4.4. Some parts of the examples have been presented before at conferences: the description of the experiments (but not the wave reproduction in CFD) in Example 1, most of the long duration stationary wave time trace results in Example 2a and the description of the experiments (but not the discussion with a focus on the scale effects in the wave crests) in Example 4. This is also indicated in the respective chapters. However, most work is new: the CFD reproduction of the green water impact results in Example 1, the long duration wave time trace results in moving frame of reference in Example 2b, the validation of breaking wave kinematics in Example 3 and the discussion on scale effects in breaking wave impacts in Example 4. Finally, Section 5 describes some conclusions and planned future work.

## 2. Aspects of Basin Wave Modelling

In order to generate basin waves, a wave board transfer function has to be derived, which depends on board type (flap, piston) and geometry. Wave board motions are generally based on linear wave theory, as derived by [14] using a method to match the wavemaker boundary conditions based on [15] and validated by [16]. In addition, a second-order correction as derived for position-controlled wave generators by [17] is usually applied to correct for discrepancies between the wave board and wave

orbital motions. In the generation of steep deterministic wave sequences at a pre-defined target location in the basin, linear wave theory may not be sufficient to calculate the required wave board motions. This can be adjusted by either applying an iterative scheme comparing and adjusting the measured time trace with the target time trace [18], or in the future by calculating the wave board motions using nonlinear wave models (CFD or potential flow methods).

With the application of nonlinear wave models, it is usually assumed that the reflections, basin modes, and other disturbances in model testing facilities do not affect the wave propagation. Alternatively, a 'digital twin' of the basin has to be modelled, including wave board(s), walls, (correct reflection and absorption of) beaches and possible other constructions such as a the basin floor bathymetry. Thus far, this has proven to be challenging without any tuning based on experimental data. The reflective properties of the beaches generally used in wave basins and the unwanted side effects of the applied type of wave generator for instance have to be known and reproduced in detail in order to validate a digital twin.

Unwanted 'basin effects' can significantly affect the generated waves, as appreciated by e.g., [8,18,19]. Their significance depends on specific test conditions such as the water depth, wave period or steepness, model scale and the properties of the basin. Especially on shallow water, low-frequency free waves and resonant basin modes can play a large role [20–22]. In large basins, wave generation and absorption can introduce small and slowly decaying return currents, due to Stokes drift [23–27] and alongshore and rip currents over the absorbing beaches [27]. These can have an effect on the measured resistance of small slow models, as well as affect the repeatability of the measured wave field over longer distances [28–30]. Such a current pattern is visualised in Figure 2. Another phenomenon that is hard to model numerically is wave breaking. The onset of breaking is sensitive to details in the wave shape and kinematics and breaking is highly nonlinear. Overturning breakers or impacts in breaking waves are hard to repeat experimentally due to instabilities associated with escaping air at high velocity [31–33]. This may also affect the overall repeatability of wave conditions in a basin. Some of these basin effects can be evaluated with numerical simulations, which make it easier to isolate and study certain phenomena. However, this first requires extensive validation of the digital twin of the basin as discussed above.

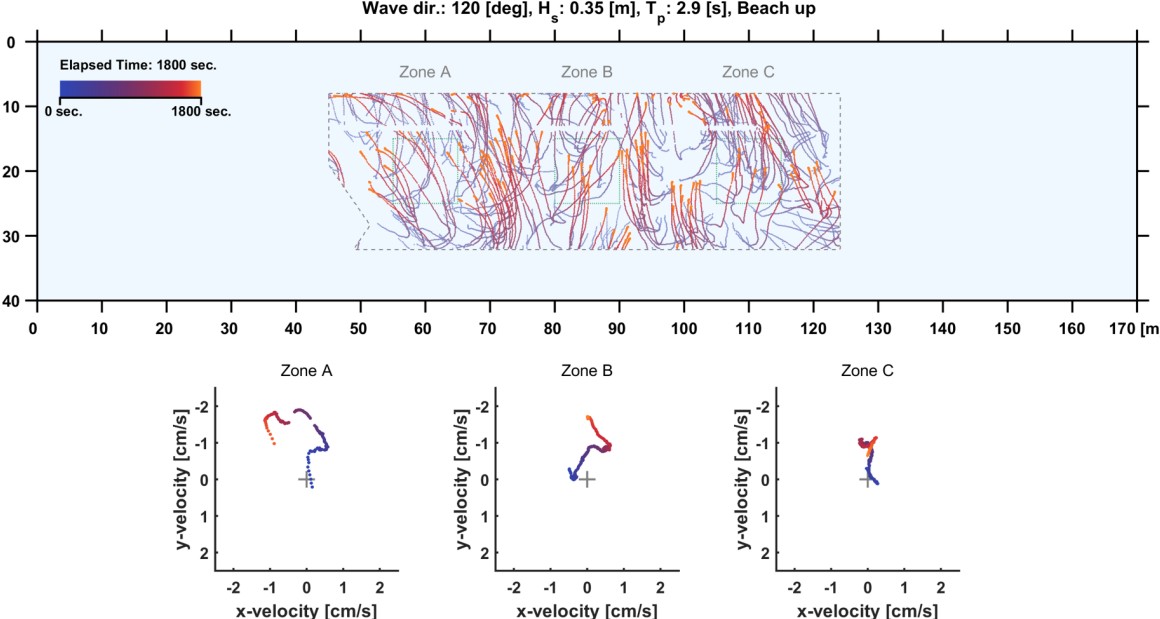

**Figure 2.** Example of a measured current pattern MARIN's Seakeeping and Manoeuvring Basin, following trackers in the 5–35 min after 30 min generation of an oblique wave (heading 120 deg so from the bottom right, long beach up)—the figure similar to those published in [27], but for a different wave condition.

If the expected basin effects are known, through basin measurements and/or simulations, they can be mitigated using adjustments to the basin (beaches, adjusting a bathymetry, etc.), active reflection compensation (ARC) or deterministic 'anti-waves' to cancel unwanted long wave reflections [34–36]. However, it can be imagined that significant structural modification of the experimental facility is not always possible. Known basin effects can also be taken into account in the analysis of basin measurements.

## 3. Aspects of Numerical Wave Modelling

Numerical wave modelling can be used for many problems in ocean engineering, but different techniques and levels of detail may be required for different applications. The driving factor for this choice is the expected degree of wave nonlinearity. For steep breaking waves for example, a CFD model is required, whereas for waves with mild nonlinearities a nonlinear potential flow may be sufficient or even an analytical second-order model. In order to obtain infragravity waves in coastal areas, a shallow-water model may be applied. In general, computational time will go down with reduced nonlinearity in the numerical models (associated with simpler model equations). In the end, ocean engineering studies aim to quantify the effect of waves on objects at sea such as ships or structures used for oil, gas, renewable energy or blue growth. This implies that numerical wave models need to be coupled to models that account for wave-structure interaction. In addition, here, the modelling choice depends on the complexity of the related physics. CFD methods can be used, but also linear or (weakly) nonlinear potential flow methods. A comparison between different numerical wave modelling approaches for a wave-in-deck event is for instance given in [37].

Wave impact simulations are generally done using CFD. Since these simulations are rather expensive, the focus is usually not on the generation of long time records but on the impact events only. The occurrence of these events can be predicted for example by screening methods [12,13,38]. Validation of wave impact simulations is done by comparing with experiments. The details of impact loads on a structure in waves depend strongly on the details of the incoming waves (as demonstrated by e.g., [39,40]). This makes it necessary to reproduce short wave sequences measured in the model test basin exactly in the CFD model. After successful validation, design variations of tested models can be considered. For proper validation, it is important to have an accurate match between the measured basin wave and the wave in the CFD model, which is not straightforward. Additionally, differences in the numerical modelling between different CFD codes can play a role in the accuracy of the results [41]. An iterative method to accomplish a good match is described in Section 4.1 and in more detail in [11]; however, other approaches towards obtaining an accurate match between basin wave and numerical wave exist; see, for instance, [42]. In Section 2, it was discussed that repeated conditions in the basin may not always reproduce exactly. In CFD, on the other hand, wave conditions with the same inflow will always result in the same crest at the target location. Thus, variations in the breaking process and crest height observed in basin measurements are not reproduced in CFD, unless the inflow conditions close to the model are accurately reproduced [11,43,44]. An example will be presented in Section 4.1.

Due to increasing computer power, longer duration long-crested irregular sea states can be generated in CFD these days. The authors in [45] demonstrate the need to include the basin boundaries in numerical reproduction of a long duration basin sea state: a three-hour sea state obtained from a wave basin measurement can be reproduced quite well in CFD up to a certain point in time. After that, reflections in the model test basin start to affect the basin measurements and the CFD results start to deviate from the basin measurements. The details of these simulations are explained in Section 4.2.

In order to validate the kinematics and the wave breaking process, experiments in breaking waves are of high interest. Some steps have been made in generating validation data for wave kinematics in CFD [46–48]. However, more work is required to further validate the breaking process and evolution of breaking waves in CFD. Some measurements of the crest kinematics in breaking waves and their comparison to CFD computations are presented in Section 4.3. Finally, the scale effects associated with breaking wave impact model tests are discussed in Section 4.4.

## 4. Examples

*4.1. Example 1: Reproducing a Single Experimental Wave Event on a Ship with Speed in CFD*

The first example of a link between experimental and numerical wave modelling concerns green water impact loads on a breakwater of a sailing containership. Similar problems were studied in the past using for instance a dam-break model on a stationary ship [49], with modifications to account for forward speed [50,51] leading to reasonably good results for impact loads in regular waves. Since then, many publications describing CFD methods to analyse extreme wave loading on stationary or moving structures in both regular and irregular waves can be found; see, e.g., [42,52–55]. In most of these publications, a large focus has been on an accurate reproduction of the basin wave, in order to make a deterministic validation of the CFD results possible. The referenced publications illustrate that multiple approaches to achieve such a reproduction exists, each with their own advantages and disadvantages. In the presented example, the approach as described in [11] has been used.

The case studied in the present example concerns green water impact loads on a breakwater on the bow of the 230 m KCS containership (this reference ship is described by [56]). Free-sailing experiments with the ship in head waves were performed in MARIN's Seakeeping and Manoeuvring Basin (SMB, [57]), measuring 170 × 40 × 5 m. These experimental data were also used by [29,30], focusing on the repeatability of the basin wave fields and the ship response. Here, the same experiments are used, but the focus is on their reproduction in CFD. A brief introduction of the set-up and test conditions is given below for convenience. The experiments were done for the Green Water Dynamics working group of Cooperative Research Ships (CRS).

The scale of the tests was one to 37.89. During the experiments, the incoming waves were recorded at several locations around the ship, as well as the ship motions, the relative wave elevation at 11 locations around the ship and at four locations on the fore deck. The breakwater (15.2 × 9.5 m) was modelled in a simplified way, using a rectangular box with 40 force panels (1.89 × 1.89 m) that fully cover the breakwater. The measurement carriage moved with a constant forward speed, and the ship was only restricted in low-frequency surge with a spring in order to prevent large speed variations. All other six-degree-of-freedom ship motions were free. Deterministic repeat tests with the model were performed using a method described in detail by [29], as well as deterministic repeat tests without the model to measure the undisturbed wave propagation at several locations on the ship centreline. The wave probe layout with and without model is shown in Figure 3.

The present example considers head waves (180 deg). The model and the carriage moved into these waves at 5 kn speed. From a three-hour full-scale wave realisation with a JONSWAP wave spectrum and $H_s$ = 6.8 m, $T_p$ = 9.7 s, $\gamma$ = 3.3, a large green water event was selected for reconstruction. The CFD analysis was performed using the CFD code ComFLOW [58–60] at full scale in single phase mode. ComFLOW explicitly reconstructs the free surface on a structured grid and integrates the free surface explicitly in time, using a variable time step to limit the Courant (CFL) number. First, the undisturbed wave run for this event was iteratively reconstructed on a 2D mesh, with 0.5 m grid resolution in both directions. To account for the constant forward speed of the vessel, a moving reference frame was employed.

As the first step in this procedure, linear dispersion theory was used to determine a first-order linear wave input at the inflow of the 2D CFD domain, based on the measured wave at W11 (target location, see Figure 3). The wave was then propagated in the CFD domain towards probes W11, W12 and W13, close to where the breakwater was located when the ship was present. The resulting wave elevations were compared to the basin measurements, and iteratively corrected until the overall match at probes W11, W12 and W13 was best. In this correction procedure, the realised wave at W11 and the corresponding measured wave at that location were compared in Fourier space and corrected. The modified realised wave was then propagated from the inflow again towards the target location. The correction based on Fourier components assumes linearity, which is gone once the realised wave starts to propagate in CFD. Hence, this iterative process does not perfectly converge, and a best match

should be selected based on visual inspection or error minimisation algorithms. Multiple parameters can be modified in the procedure: the location of the inlet is free to be chosen, as well as the time interval around the selected wave and the space windowing around the selected time interval. Hereby, the latter two are to be selected such that each of the wave components is allowed sufficient time to propagate from the inflow location to the target location.

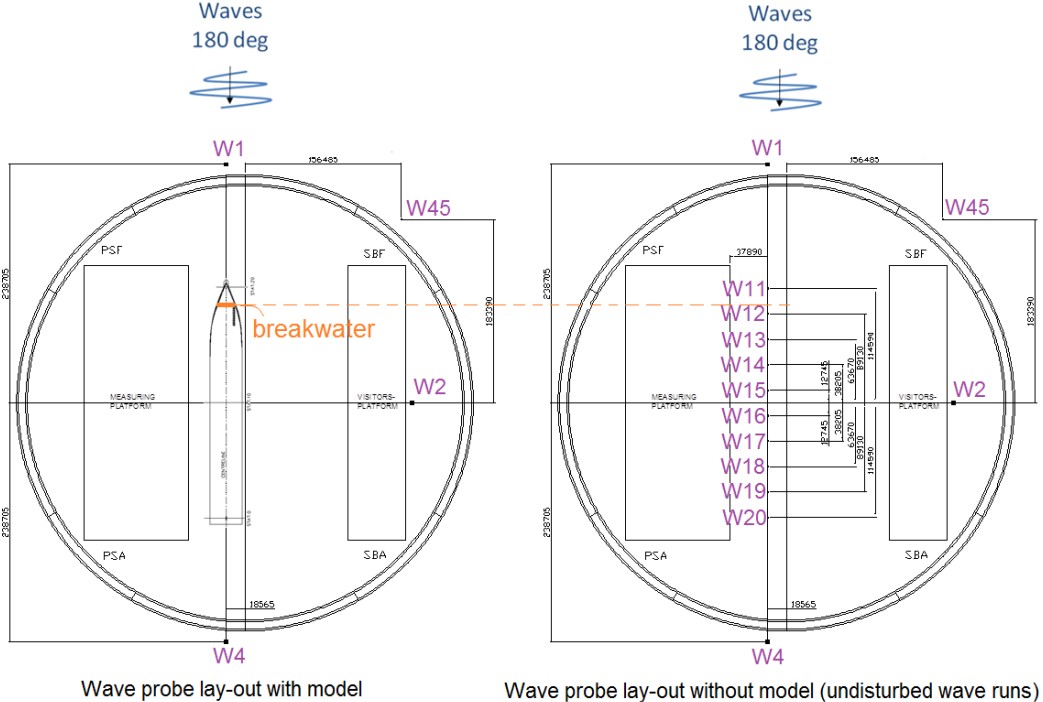

**Figure 3.** Wave probe layout under the (circular) carriage with and without model, moving with a fixed speed against the 180 deg waves (where W# = wave probe number).

The 'best' 2D wave realisation from this procedure for the selected green water event is shown in Figure 4. It was selected based on several criteria: the water elevation at the target location (emphasising on the crest), the derivative of the water elevation (rise time of the water) and the propagation of the realised wave in time. Once the best iteration was selected, the wave kinematics corresponding to the small time interval of interest were stored. Interpolating these kinematics on the inlet of a 3D domain made it possible to run only this small time interval in 3D. The 3D domain included the ship. The six degrees of freedom ship motions from the experiments were prescribed in the calculations. Around the breakwater, the computational grid was locally refined to 0.125 m in all three directions. Outside the area of interest, the grid was coarsened to 8 m, resulting in a grid of 27.2 million grid cells. Due to the single phase mode, only the wetted grid cells were solved, hence the number of solved grid cells decreased drastically and varied from 2.4 to 6.9 million. During the green water impact, the time step size decreased to $2.2 \times 10^{-4}$ s in order to capture this highly nonlinear event accurately. The total amount of Central Processing Unit (CPU) hours required for the simulation was 4200 h (wall clock time of 7.3 days on a single node of 24 cores). The result of the simulations is shown in Figure 5. A very good match between the experiments and the CFD results was achieved, both for the undisturbed incoming wave and the local loads on the breakwater.

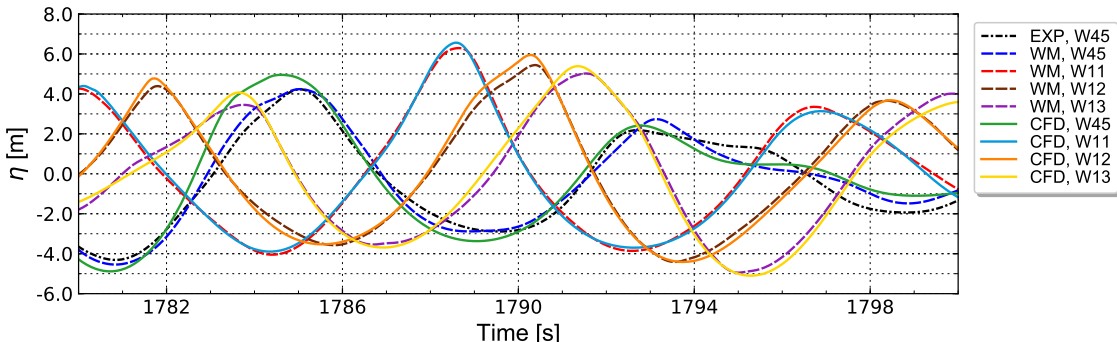

**Figure 4.** The best realised reconstructed undisturbed wave compared to the measured undisturbed wave, where 'WM' indicates the waves measured during the undisturbed wave experiments at probes W45, W11, W12 and W13, 'CFD' the undisturbed numerical reconstructions at the same locations and 'Exp' the wave measured during the experiment with the model at probe W45 (for reference). Figure based on the same data as a figure in [44].

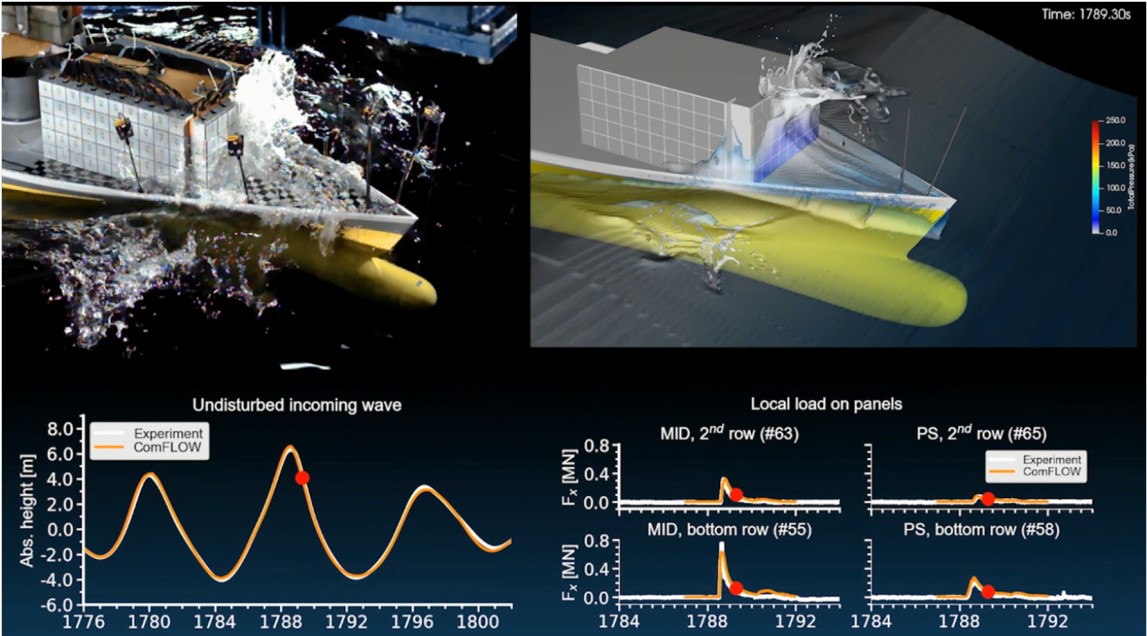

**Figure 5.** Synchronised green water event during the experiment (**left**) and in the CFD calculation (**right**), with the associated undisturbed wave elevation at the bow location and forces on four of the breakwater panels. Figure based on the same data as a figure in [44].

During the basin testing, deterministic repeat tests were performed and results showed that the variability in the basin waves as well as the ship motions and loads on the breakwater increased with distance from the wave generator [29,30]. The wave generator flap motions are well repeatable, as is the synchronisation with the carriage motions. However, as the wave propagates, small wave-induced currents or free-surface instabilities in (nearly) breaking waves can increase the wave variability with propagation distance (as discussed in the introduction and [27]). Figure 6 shows results for ten deterministic repeats tests, and shows that the variability in the impact forces typically is a lot larger than that in the incoming waves, confirming the large nonlinearity in the problem. However, by accurately matching the vessel motions and the incoming wave conditions at a location close to the impact, the CFD results showed that most of the variability in the output can be eliminated by removing the variability in the input. A certain amount of variability in the wave loading will however always remain—for instance, due to free-surface instabilities caused by escaping or entrapped air [31,33]. For this specific example, however, variabilities due to two-phase phenomena seem to be

small, as the impacting water was mostly running up from below against the breakwater (see Figure 5) and no significant amount of air was enclosed during the impact. If enclosed or escaping air in a high-velocity jet plays a role in an impact, it is expected that the variability in the loads is higher than can be derived from the waves and motions, and two-phase calculations may be required, as further explained in Example 4 in Section 4.4. Some variations in the CFD inlet conditions (small amplitude or timing variations) may be considered in order to evaluate the sensitivity of the output.

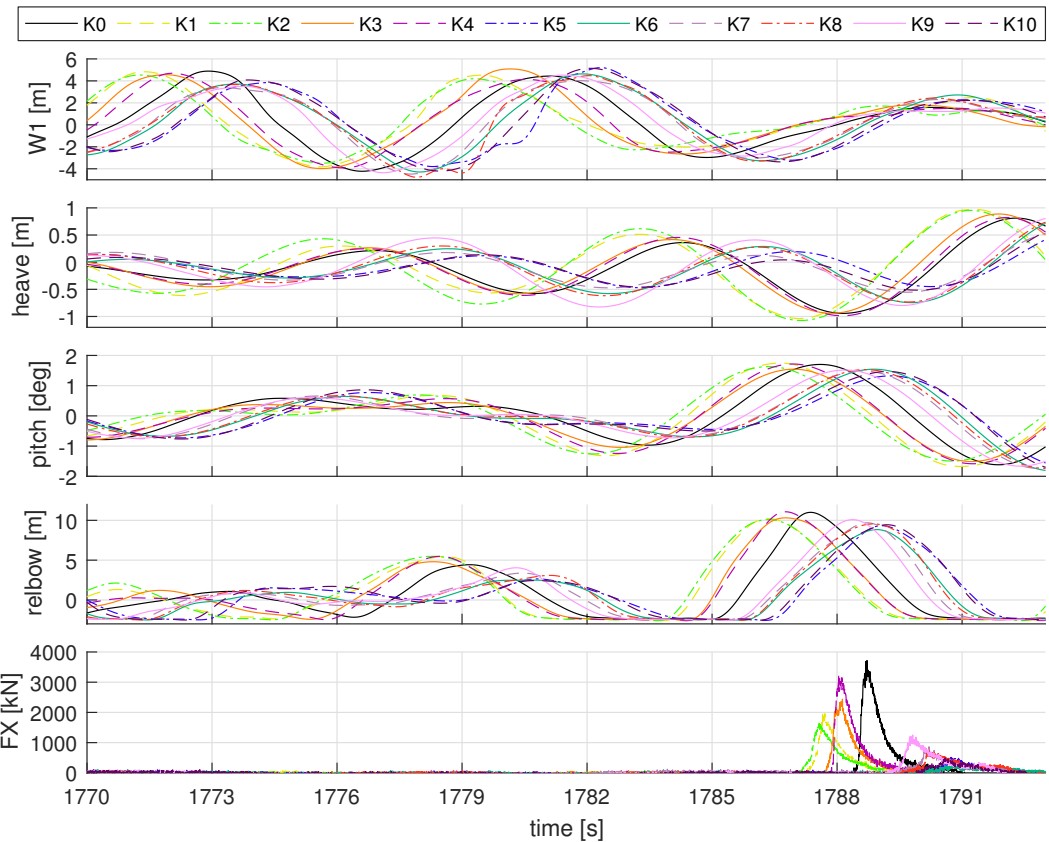

**Figure 6.** Deterministic repeat runs using synchronised wave generator flap motions with measurement carriage motions, incoming wave (W1, where the strange behavior on some steep wave slopes is caused by the way the acoustic wave probe records a breaking wave), ship motions heave and pitch, relative wave elevation on the bow (relbow) and integrated force on the breakwater (FX) around 4700 m full-scale from the wave generator—similar to a figure in [30], but for a different event.

### 4.2. Example 2: Reproducing Model Tests in Long Irregular Wave Sequences in CFD

Example 1 showed the detailed reproduction of wave event with a duration of a few seconds from the basin. This is very useful—for instance, for validation of the impact loading in CFD. However, for other applications, the reproduction of longer wave time traces is required—for instance, in order to validate motion response statistics or to screen the wave realisation for extreme events. The present section discusses numerical reproduction of a longer wave realisation.

Nonlinear wave generation for longer durations is possible in nonlinear potential flow methods, as demonstrated by, for instance, [61,62]. However, such methods typically break down when waves start breaking (or rely on empirical dissipation models), which affects the tail of the wave crest distribution. CFD methods do not have this limitation, but they are computationally expensive. For instance, the authors in [63,64] show that it is possible to model a floating platform in a three-hour wave realisation in CFD, leading to good results for the motion response functions. Long-duration wave reproduction in CFD is also discussed in the present example, with an emphasis on accurate deterministic numerical reproduction of the wave crests, considering possible unwanted basin and

numerical effects. This provides more insight in the potential as well as the limitations of both types of modelling. Such long-duration wave reproduction is discussed in Section 4.2.1 for an earth-fixed location and in Section 4.2.2 for a moving frame of reference.

### 4.2.1. Example 2a: Long Duration Wave Reproduction at an Earth-Fixed Location

The present example discusses work done within the WiFi JIP (Wave Impacts on FIxed offshore wind turbines Joint Industry Project). This work included results for long duration experimental wave reconstruction in CFD at a fixed location, which were presented earlier in [45]. As these results form a good introduction for the work on long duration wave time traces in a moving frame of reference (as discussed in the next Section), they are summarised here. Note that, within the WiFi JIP, short duration wave matching was also applied, similar to Example 1; see, for instance, [65].

Steep and breaking waves were generated in MARIN's Shallow Water Basin (SWB), see Figures 7 and 8. The basin measures $200 \times 16$ m, and for these tests the water depth was 0.75 m at a scale of 1:40. The purpose of the tests was to measure (breaking) wave loads on fixed offshore wind turbines to provide validation material for simulation models. 3D CFD models are very capable of predicting these loads [65], but they are expensive. Therefore, an alternative was investigated: can the long-crested wave kinematics resulting from a 2D CFD model be used in a Morison model to estimate the wave loading? To test this principle, a CFD reproduction of the measured basin waves with a duration in the order of 3 h full scale is needed. A wave realisation with a JONSWAP wave spectrum and model-scale properties $H_s = 0.25$ m, $T_p = 2.06$ s, $\gamma = 3.3$ was selected ($H_s = 10$ m and $T_p = 13$ s on full-scale). This wave signal was measured at two locations in the basin, with a distance of 10.0 m model-scale in between. The first measurement location was used to derive a boundary condition for the CFD model. The measured wave elevation was used to compute the volume fraction of the boundary cells and second-order wave theory [66] was used to derive the water particle kinematics on the CFD boundary. The wave was then propagated through the CFD domain and compared to the experiments at the second measurement location. The CFD model ReFRESCO was used [67].

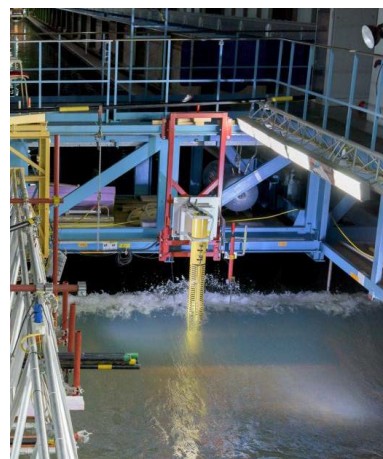

(**a**) Experimental set-up.

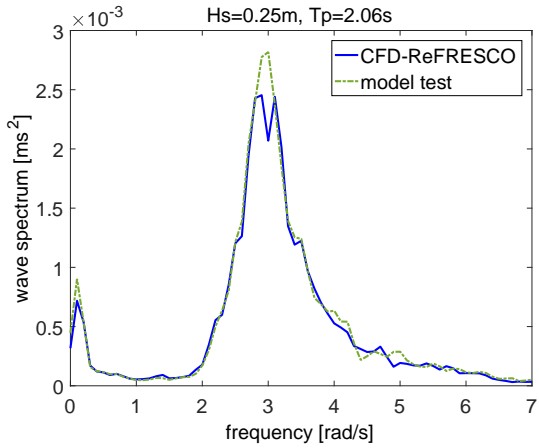

(**b**) Model-scale wave spectrum.

**Figure 7.** Wave spectrum derived from wave measurements and CFD calculations on steep and breaking waves approaching a fixed monopile. Figure based on the same data as a figure in [45].

Figure 7 shows the wave spectrum derived from the basin and CFD wave signals. A good agreement was obtained, with only a small discrepancy at the peak of the spectrum, probably due to reflections in the basin (as discussed later). Clearly visible is the low-frequency wave content due to infragravity waves in shallow water. When the wave signals were compared, however, it was observed that the match was very good in the initial stages of the simulation (Figure 9a). but became less good later on (Figure 9b). This phenomenon was further investigated, and it can be attributed to basin

reflections. The SWB is fitted with a parabolic beach to dissipate the waves. This works well for short waves, but long waves are partially reflected. This phenomenon was extensively investigated in the HAWAI JIP [68] and is common to any experimental wave basin.

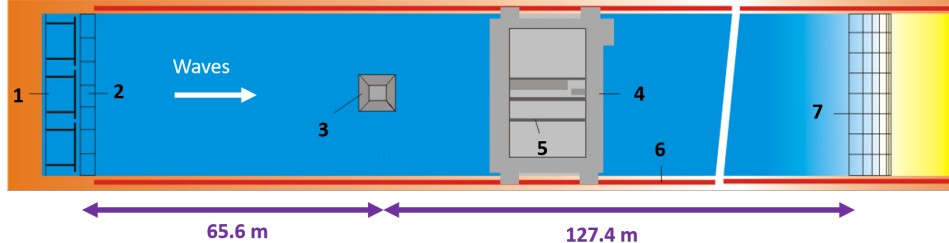

**Figure 8.** Schematic overview of the shallow water basin. 1: Piston-type wave generator. 2: Strips to reduce cross-wave resonance. 3: Pit where wind turbine was mounted. 4: Measurement carriage. 5: Sub carriage. 6: Rails for carriage. 7: Damping beach.

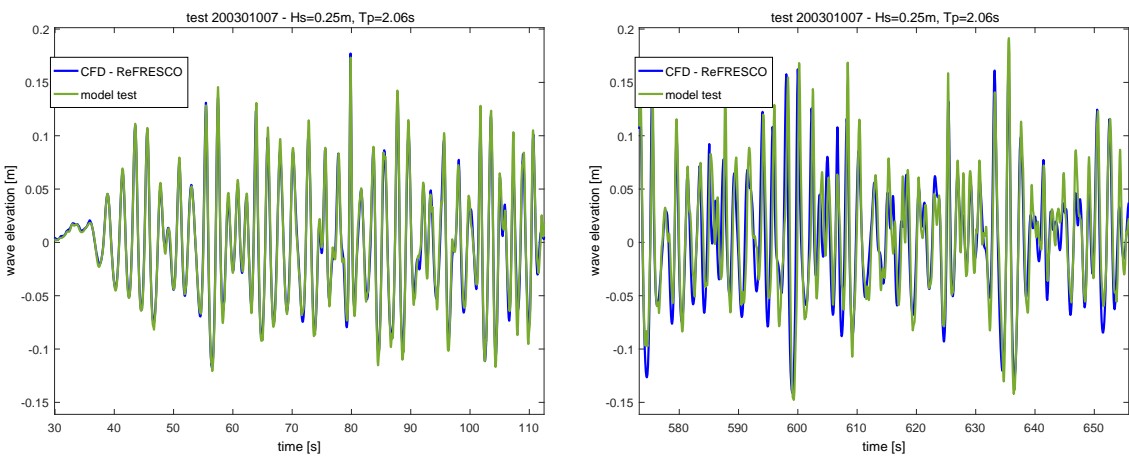

(**a**) 3–12 min full-scale from start simulation.

(**b**) 60–70 min full-scale from start simulation.

**Figure 9.** Measured ('model test') and simulated ('CFD - ReFRESCO') wave elevation for two parts of the time trace. Figure based on the same data as a figure in [45].

The 'effective' length of the basin (wave generator to beach) is 193 m; see Figure 8. Based on the shallow-water group speed (2.71 m/s for a water depth of 0.75 m), the time it takes for the first long-wave reflections to propagate through the basin and reach the set-up can roughly be estimated (the shortest waves in the spectrum will propagate at a somewhat smaller speed). Figure 10 shows the cross-correlation between the measured and simulated wave. On the horizontal axis, the time (from the start of the measurement) used to compute the correlation is shown. After arrival of the first waves, the correlation is very high (>0.98) for approximately 100 s model-scale. The instances in time where the correlation suddenly drops to a lower value can be identified as the moments in time when reflections arrive at the measurement location. First, reflections from the beach, then the beach reflections reflecting on the wave generator, etc. Towards the end of the simulation, the correlation has dropped to a value of 0.9. The consequence of this is that a deterministic comparison between the measured and simulated wave signal is unfortunately not possible for the full 3.5 h duration. Based on these observations, the following conclusions seem justified:

- A deterministic comparison of long time traces is only possible when basin reflections have not arrived at the measurement setup. This happens already quite quickly in a 3.5 h model test (in this case, only 100 s model scales or 600 s full-scales are free of reflections, 5% of total measurement time).

- The correlation between measured and simulated waves at the position of the wind turbine is
  very good prior to the arrival of basin reflections. The correlation deteriorates after arrival of
  wave reflections. Improvement can only be obtained by modelling basin effects (beach reflections)
  in the numerical model.

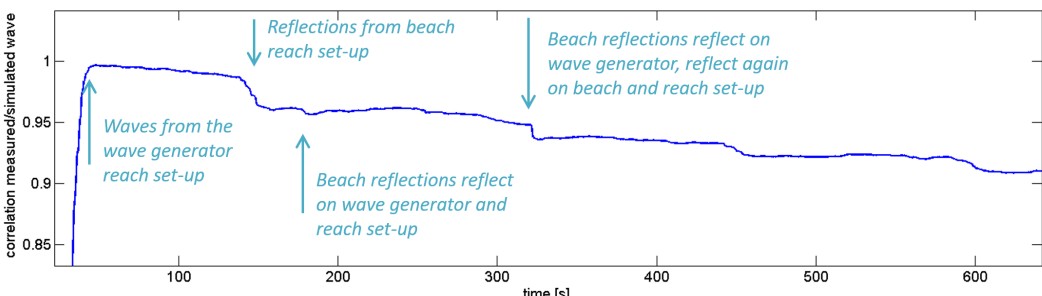

**Figure 10.** Deteriorating correlation between measured and simulated wave elevation at the location
of the wind turbine linked against arrival of wave (reflections). Figure based on the same data as a
figure in [45].

### 4.2.2. Example 2b: Long Duration Wave Reproduction at Forward Speed

For the present example, undisturbed wave data from the same experimental campaign as
discussed in Section 4.1 were used. These results were not published before. A long undisturbed
wave measurement was reproduced in CFD. Figure 3 on the right again depicts the experimental
set-up. Contrary to Example 1, here the purpose of the reproduction concerned the validation of the
vessel response prediction using CFD (in particular, the relative wave elevation at the bow), instead
of a prediction of green water loads. Therefore, wave probe W15 (located close to where the centre
of gravity of the vessel would be) was selected as target probe instead of wave probe W11 (located
closest to the green water impact location). Additionally, instead of focussing on the (very) accurate
reproduction of just one wave crest using an iterative procedure as was done in Example 1, here a
complete test of 30 min (full scale) was reproduced using the same procedure as discussed for the
earth-fixed wave reproduction example in Section 4.2.1. The inflow of the CFD domain was chosen
to coincide with the location of measurement probe W1, and the measured data at this location was
used to compute the volume fraction and water particle kinematics used as inflow boundary condition
to the CFD model. Again, second-order wave theory [66] was used to compute the inflow boundary
water particles, and, for the CFD simulations, ComFLOW was used.

Figure 11 shows the measured and simulated wave elevation for the complete duration of the
reproduction. The wave had a full scale $H_s$ of 6.8 m and $T_p$ of 9.7 s. As the wave was measured
(and reproduced) at a forward speed of 5 kn, the reference probe reached the other end of the
basin before significant basin reflections returned at its location, so the influence of the reflections is
negligible (contrary to the earth-fixed example). A good match is therefore achieved throughout the
whole time trace. Figure 12a shows the distribution of the crests for the measured and simulated wave,
and confirms the quality of the match. It should hereby be stressed that, even though the distributions
match very well, the individual measured and simulated crests at the same probability level in this
figure do not necessarily correspond to the same individual events.

Taking one step further in quantifying the quality of the achieved match, a deterministic match
between the measured and simulated crests was made. Hereto, the crests from the experiments and
from the numerical simulations were individually numbered and matched, as shown in Figure 11.
This individual numbering of the separate crests makes it possible to correlate the individual crests,
instead of comparing them based on their statistical distributions. Figure 12b shows the resulting
correlation. A mean error (measured crest heights minus corresponding simulated crest heights) of
0.056 m was found together with a Root-Mean-Square (RMS) error of 0.460 m (6.7% of $H_s$), confirming
that the reproduction also deterministically is of high quality.

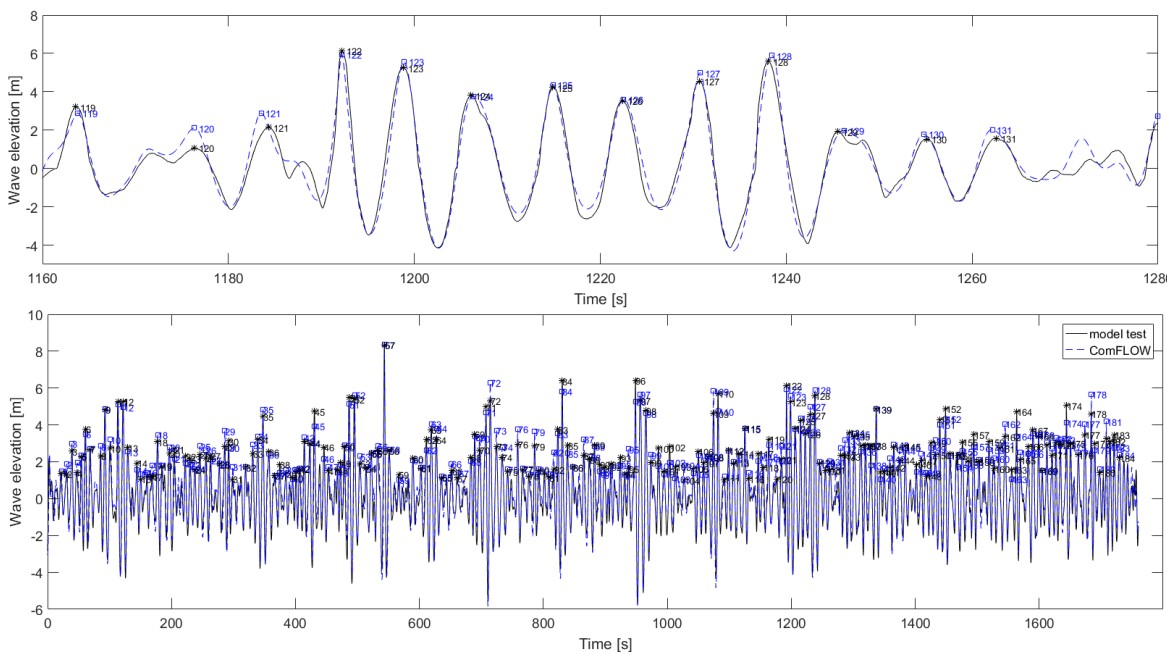

**Figure 11.** Measured and simulated wave elevation at probe W15.

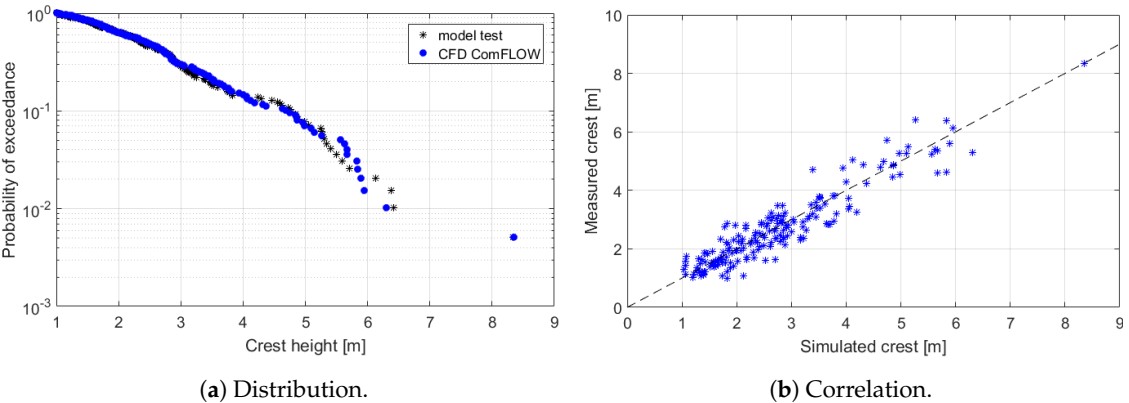

(**a**) Distribution.

(**b**) Correlation.

**Figure 12.** Measured and simulated wave crest distribution and individual crest correlation for measured and simulated wave elevation signal, both at probe W15.

### 4.3. Example 3: Validating Numerical Wave Kinematics Using PIV Measurements

Breaking waves may exert large loads and stresses on offshore structures. With regard to structural survivability, breaking wave impact has been identified as one of the most dangerous conditions [69–71]. Since the kinematics of the water under the breaking crest determine the magnitude of the loads to a large extent, many experimental and numerical works have focused on this topic. As it allows a non-intrusive investigation of the velocity field underneath breaking waves at laboratory scale, optical techniques such as Particle Image Velocimetry (PIV) have been popular experimental techniques [72–74]. However, both measurements of the breaking wave kinematics and validation of CFD tools in this complex problem require further research. In [72], kinematics under a breaking wave were calculated with a two-phase CFD code and the results were compared to PIV measurements. It was observed that the kinematics close to the wave crest were underestimated in the simulations. Results of Large Eddy Simulations (LES) for plunging wave breaking were compared to PIV results in [75], where a good agreement was observed in the flow trends in mean velocity and vorticity.

In the present example, the kinematics under breaking waves were observed using both experimental and numerical methods. Spilling and plunging type breakers were created using dispersive focusing in a laboratory flume, and their kinematics were measured using PIV. The same

waves were later recreated in CFD code ComFLOW, using the same iterative scheme described in Section 4.1.

### 4.3.1. PIV Measurements

The PIV measurements were conducted in a dedicated test facility: the MARIN Research Flume (RF). This flume measures 5.5 × 0.3 m, with a water depth of 0.5 m (Figure 13). At one end of the flume, a single flap-type wave generator is installed. At the other end, a beach and a foam structure minimise reflected waves. A first wave measurement campaign was performed in 2015 using a low-speed PIV system. Results were described in [47]. With this low-speed system, average correlations between different repeat tests were used to obtain the kinematics in the wave crests. This was only possible by assuming that the waves were almost perfectly repeatable. As this campaign only studied one non-violent spilling breaker, this assumption was not perfect but acceptable. However, in order to obtain wave kinematics in violently breaking waves, high-speed PIV measurements are required. Such a system was used for time-resolved measurements in the RF in 2019, for nine different spilling and overturning breakers. The present example discusses results from these new tests for the first time. Two 4-megapixel cameras were used simultaneously in a 2 × 2D-PIV single frame set-up—one recording at 800 and the other at 1400 frames per second. A double pulse laser was used (30 mJ at 1000 Hz). In order to synchronise the wave generation with the PIV measurements, the start pulse of the laser was used to trigger the wave generator. This synchronisation in combination with the high-speed PIV system made it possible to obtain the crest kinematics from a single test and to study the reproducibility of the breaking waves by comparing different repeat tests.

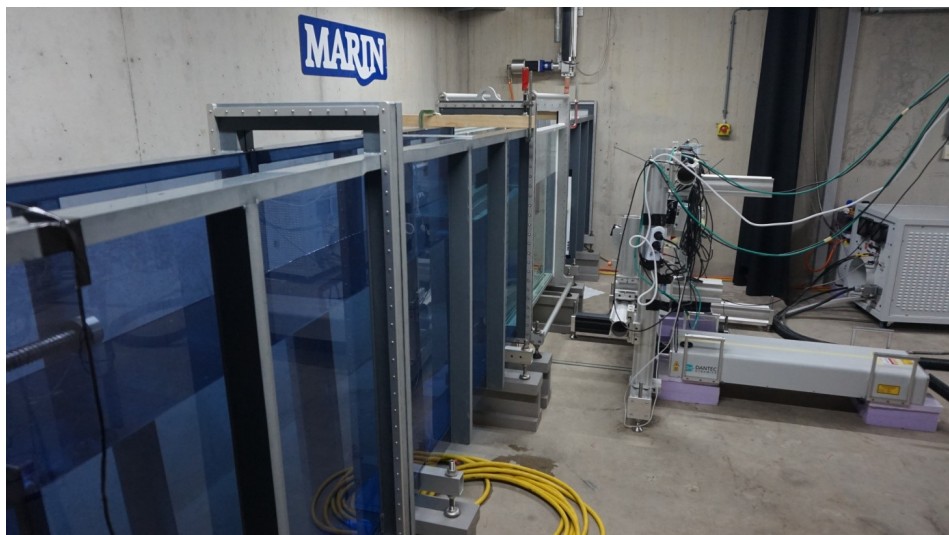

**Figure 13.** MARIN Research Flume with a breaking wave segment.

The main challenge in conducting wave kinematics measurements is to properly capture the particle velocities in the turbulent air–water mixture in breaking waves. This mixture results in reflections of the laser sheet. For that reason, fluorescent seeding was tested. However, due to problems with the dispersion of the fluorescent particles close to the free surface, the best results were obtained with polyamide. Prior to the velocity determination, a boundary detection has to be performed. An automated process was defined in which the surface was tracked over time and the intensity in bright areas was reduced. After this pre-processing, the PIV correlation was performed resulting in instantaneous velocity fields. Figure 14 shows a result of the PIV recording.

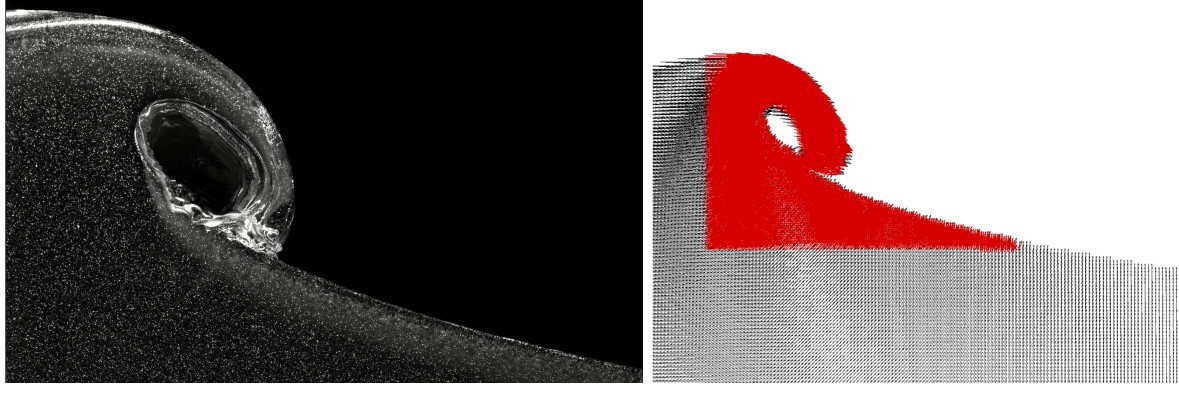

(**a**) Recording single camera.　　　　　　　(**b**) Resulting vectors two cameras combined.

**Figure 14.** Example of PIV recording.

### 4.3.2. CFD Simulations

In order to compare the kinematics from the CFD simulations with the PIV measurements, the focusing and the subsequent breaking events first need to be reproduced in the CFD domain. This was achieved again through a scheme whereby the wave at the inflow was modified iteratively according to the difference between the simulated and measured wave trains at a target location; see [48] and Section 4.1 for the details of this approach. The simulations were carried out in a computational domain shown in Figure 15. Through local grid refinement, the finest grid was generated only in the area of interest where the waves propagate from the inflow and break, which is denoted as *Refinement 3* in the figure. Gradually coarser grids were generated around the finest grid, and the resolutions and sizes of each region are also given in Figure 15. In all the refinement regions, uniform grid resolution was adopted. At the end of the domain, an absorbing boundary condition [76] was used, and care was taken to ensure that there were no reflected waves in the area of interest.

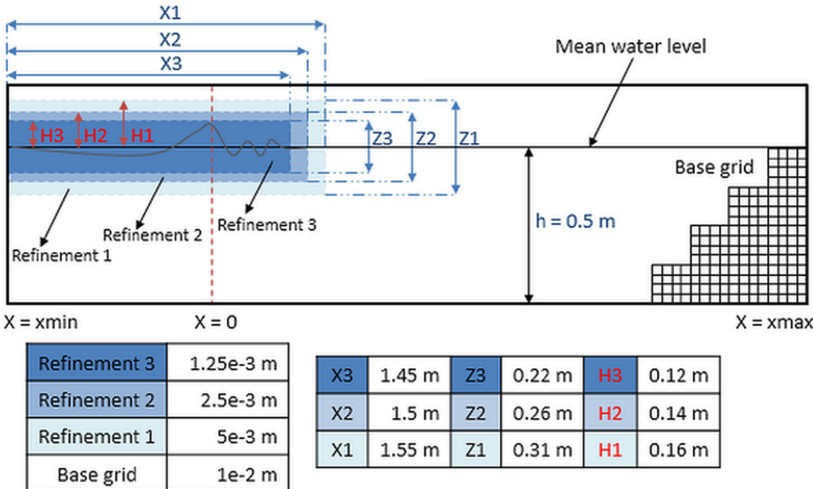

| Refinement 3 | 1.25e-3 m |
| --- | --- |
| Refinement 2 | 2.5e-3 m |
| Refinement 1 | 5e-3 m |
| Base grid | 1e-2 m |

| | | | | | |
| --- | --- | --- | --- | --- | --- |
| X3 | 1.45 m | Z3 | 0.22 m | H3 | 0.12 m |
| X2 | 1.5 m | Z2 | 0.26 m | H2 | 0.14 m |
| X1 | 1.55 m | Z1 | 0.31 m | H1 | 0.16 m |

**Figure 15.** The computational domain with three local grid refinement regions around the area of interest. The location x = 0 m was considered as the target location where the focusing events from the measurements were reproduced in the computational domain. Note that the base grid covers the entire domain.

For the sake of brevity, only the results from one plunging wave are shown here. Figure 16 illustrates the wave elevation histories from the measurement and CFD at three locations; the target location at x = 0 m and 130 mm before and after the target location. Reasonable matches were obtained overall, and especially in front of the crest at the target location the wave elevation from the simulation follow that from the measurement very closely. Furthermore, the measured crest height is very

accurately reproduced in the CFD at the target location. Note that, at 130 mm after the target point, an unexpected wave profile was recorded by the resistance type wave probe. Such issues arise with this type of probes when the breaking is violent having complex substructures with air–water mixture.

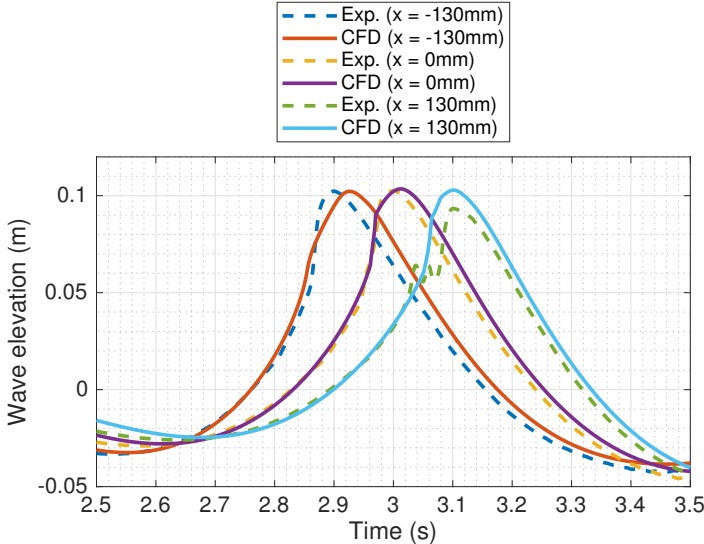

**Figure 16.** Wave elevation histories for the plunging breaker at three locations; the target location at x = 0 m and 130 mm before and after the target location.

Figure 17 shows the horizontal velocity from the measurement and simulation. Contour plots of the horizontal velocity are shown in Figure 17a at four different instances around the moment of breaking. The velocity profiles were extracted at the same instances at the x = 0 m location denoted by the dashed line in Figure 17a, and plotted in Figure 17b. It can be observed that there is overall a reasonable match between the measurement and simulation. The kinematics from the PIV are typically larger than those from the CFD simulation, especially in the overturning crest area. The collapsing water column is thicker in size and the breaking occurs in a more violent fashion in the measurement compared to the simulation. Part of this difference comes from the fact that the simulation was carried out in single phase without a turbulence model. However, with regard to the horizontal kinematics, the CFD simulation overall produces similar results to the experiment.

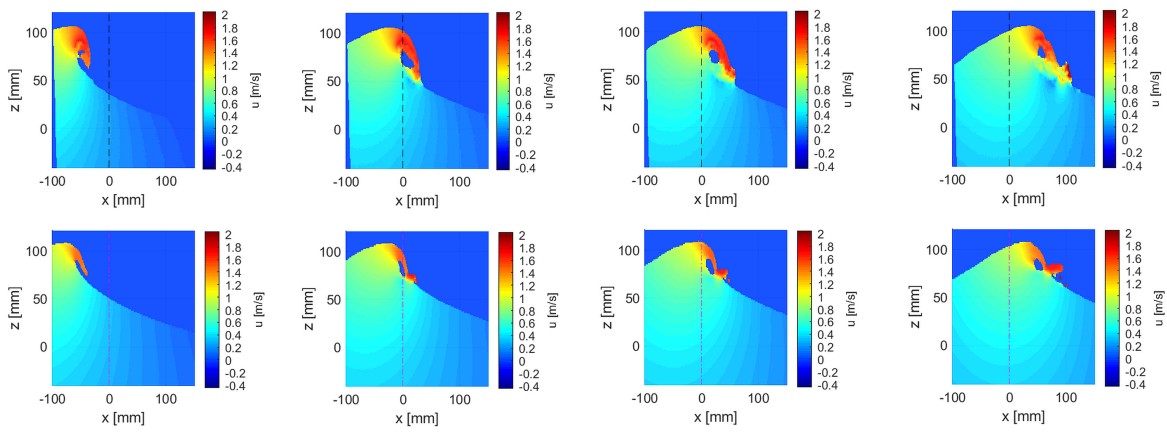

(**a**) Contour plots of the horizontal velocity around the instant of breaking. The top row shows the results from the PIV measurement, and the bottom row from the CFD simulation.

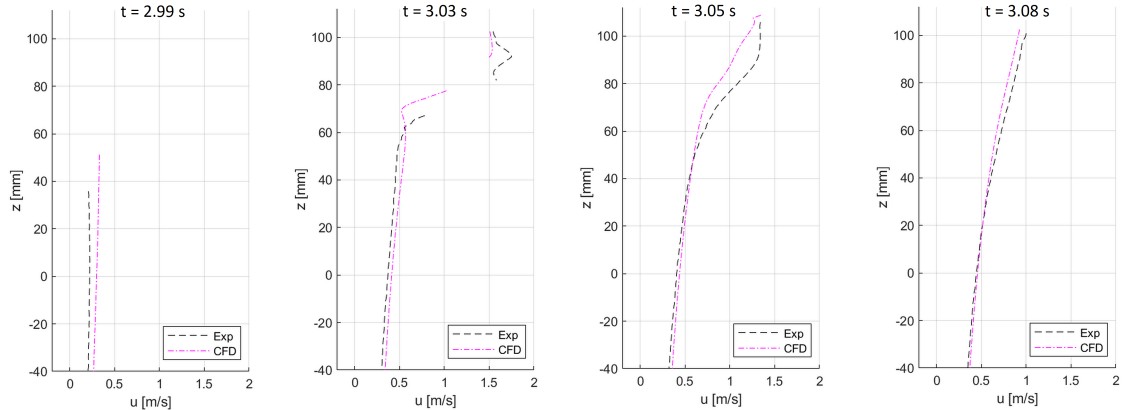

(**b**) Horizontal velocity profiles around the instant of breaking.

**Figure 17.** Comparison of the horizontal kinematics for the plunging breaker. From left to right, plots were produced at different instances before and after breaking. Note that contour plots and profile plots are produced at the same instant in time. The velocity profiles were extracted at the x = 0 m location denoted by the dashed line in Figure 17a.

### 4.4. Example 4: Scale Effects in Breaking Waves

In order to quantify the loading due to breaking waves, model tests are currently the preferred option. Rapid advances are made in the numerical modelling of wall impacts due to sloshing, including air entrapments (see for instance [31,32]). However, it is not yet clear whether two-phase modelling is necessary for all types of wave impacts and what this means for the quality of experimental scale modelling. It is suspected that scale effects lead to an overestimation of the full-scale loading, but there are questions on how conservative such load measurements are. To understand how realistic model testing is in this respect, it is necessary to quantify scale effects in the measured loading due to breaking waves as well as to investigate to what extent entrapped air in the wave and during the wave impact is involved. This knowledge will also provide guidance for numerical modelling (model- or full-scale, one or two phases). In [77], the scale effects on bubble formation in breaking waves are analysed. In the study, the same mechanisms and type of bubbles are observed in breaking ocean waves and scaled breaking waves. In [78], scale effects in hydraulic engineering models are reviewed, including the effect of surface tension on bubble formations in breaking waves. However, details on scale effects on the magnitude of the bubble formation are not discussed.

In order to shed more light on these scale effects, wave-in-deck model tests were carried out for the BreaKin JIP in MARIN's Depressurised Wave Basin (DWB, measuring 240 × 18 × 8 m) at two scales (25 and 50) in atmospheric and depressurised condition. The wave-in-deck impact loads

and scale effects thereon were presented in [33]. The scale effects on the details of the undisturbed wave conditions calibrated for these tests were not discussed in detail in that publication. These are discussed here. Five focused wave packets were generated with varying steepness at the focussing point. Figure 18 shows the measured wave elevation of the five focused waves and their steepness (time derivative of the wave elevation). The figure shows that Wave 2 is the steepest focused wave and Wave 3 the least steep condition. Wave 4 shows a small double peak in the steepness curve indicating the wave is already breaking at the target location.

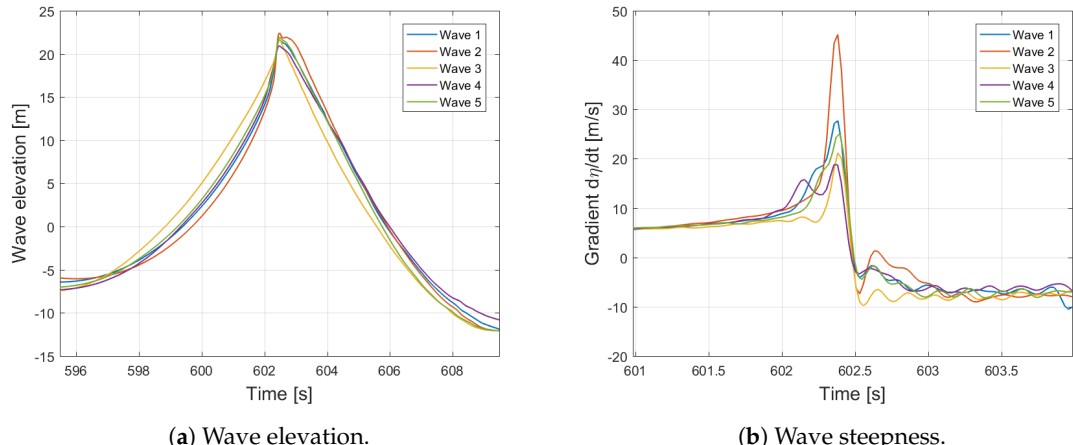

(**a**) Wave elevation.　　　　　　　　　　　(**b**) Wave steepness.

**Figure 18.** Five focused waves and their steepness calibrated for the BreaKin JIP. Figure based on the same data as a figure in [33].

Each focused wave was repeated 10 times to analyse the repeatability of the waves without model present in the basin. The 10 repeat measurements of Wave 1 are shown in Figure 19. In general, these focussed wave conditions repeat very well, in particular in overall shape and phasing. The repeatability of such single wave events is better than of long irregular wave realisations such as discussed in Section 4.1 (residual wave-induced currents do not have time to build up). However, when zooming in on the crest of the wave, Figure 19 shows variations of approximately 0.5 m full scale in crest height can be observed for all wave conditions.

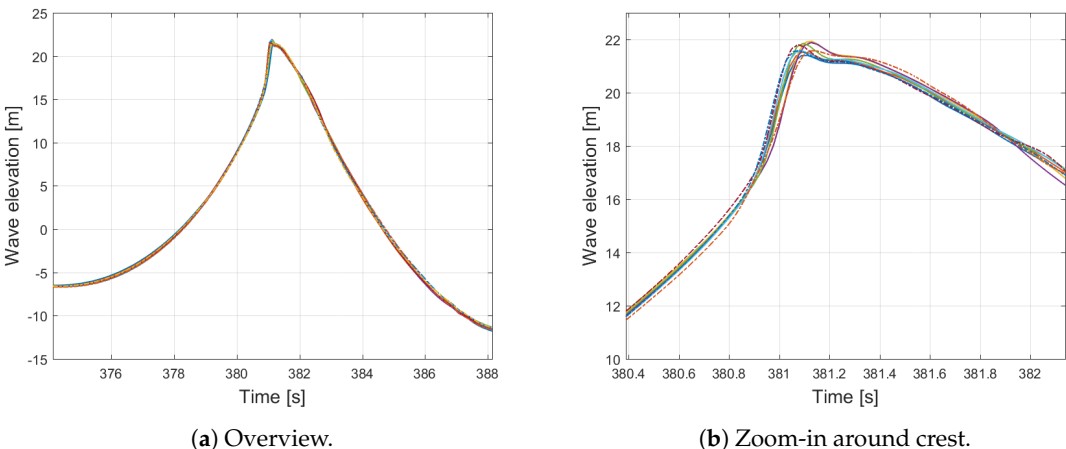

(**a**) Overview.　　　　　　　　　　　　(**b**) Zoom-in around crest.

**Figure 19.** Repeatability of focused Wave 1 in atmospheric condition. Figure based on the same data as a figure in [33].

To see whether a change in ambient pressure has an effect on the undisturbed wave, each wave was measured 10 times in atmospheric condition as well as 10 times in depressurised condition (50 mbar) without model in basin. In Figure 20, the 10 repeat measurements of Wave 1 in atmospheric condition are shown, together with one measurement in depressurised conditions (red line). As can be seen in the figures, the red line falls nicely in between the 10 repeat measurements in atmospheric conditions. Thus, it appears that the depressurisation has no significant effect on the shape of the undisturbed wave.

The initial wave conditions described above where calibrated during the first test campaign at scale 25. For the second test campaign at scale 50, the attempt was made to reproduce the same wave conditions. The distance to the wave maker as well as the input for the wave conditions were scaled according to Froude scaling. However, the wave maker itself (hinge depth) was not scaled. Therefore, the generated waves at scale 50 differ from the waves generated at the wave maker at scale 25. In addition, wave propagation, nonlinear interaction and breaking processes may also change. Thus, the scaled wave conditions were first measured once and then tuned to match the scale 25 wave conditions as close as possible. The tuning was mainly based on shifting the target focusing point and reducing the crest height with the objective to match the front slope and steepness of the individual waves.

As for the waves calibrated at scale 25, the waves also calibrated at scale 50 were measured 10 times with 15 min waiting time between the measurements. The repeat measurements were carried out in atmospheric condition as well as depressurised condition. In Figure 21, the 10 repeat measurements of Wave 1 are shown at scale 25 and scale 50. In the figure, only Wave 1 is shown; however, the trend shown in the figure is representative for all waves. At scale 25, the wave conditions repeated well with some deviations in the crest (see left figure in Figure 21). At scale 50, in general, the overall wave conditions also repeat well, but the deviation in the crests is clearly larger at scale 50 than at scale 25 (see right figure in Figure 21). This larger deviation in the crest of the repeat measurements is both observed in atmospheric as well as depressurised condition for all five wave conditions.

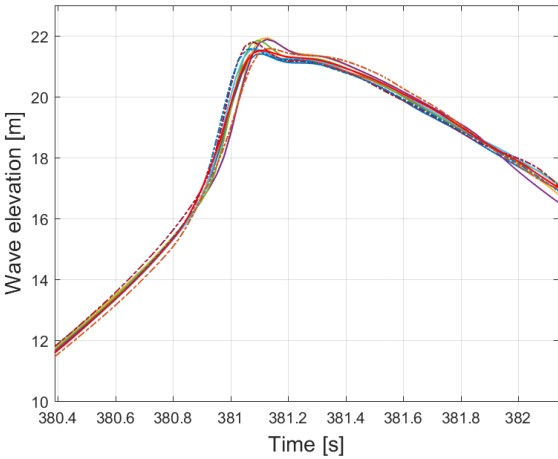

**Figure 20.** Focused Wave 1 in atmospheric and depressurised condition (50 mbar, red line). Figure based on the same data as a figure in [33].

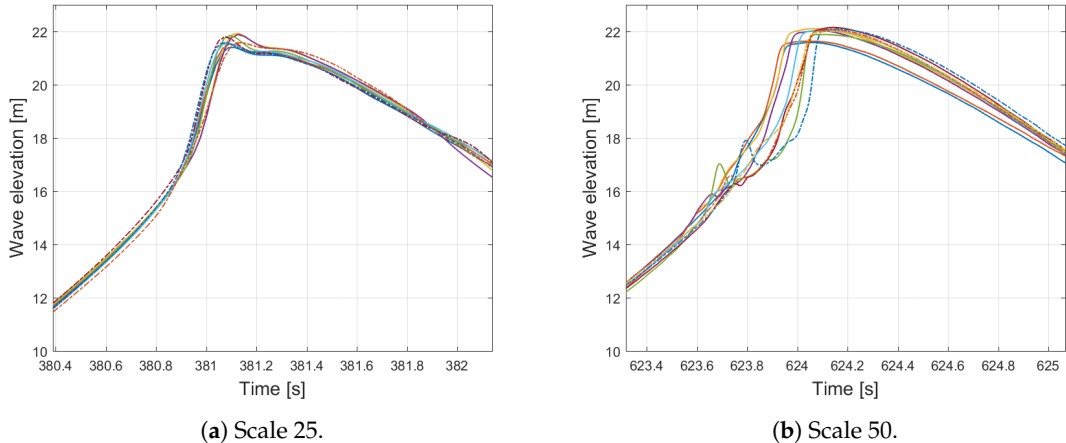

(**a**) Scale 25.

(**b**) Scale 50.

**Figure 21.** Repeatability of focused Wave 1 at two scales—both in atmospheric conditions.

In addition to the larger deviations in the crest between repeat measurements, differences in breaking process were also observed between wave measurements in atmospheric and depressurised condition at scale 50. Especially for Wave 4, which is a fully breaking wave at the target location, the deviations are large. The repeat measurements in atmospheric condition and depressurised condition of Wave 4 are shown in Figure 22. In atmospheric conditions (Figure 22a), it appears that the wave is fully overturning as it reaches the target location. In depressurised conditions (Figure 22b), the wave breaking is less pronounced and the wave is less steep.

The differences observed between the two scales and between atmospheric and depressurised condition at scale 50 are most likely related to changes in surface instabilities due to scaling and changes in density ratio of water and air in depressurised condition. In the ongoing Sling project, the modelling of surface instabilities is currently studied. Part of the scope of the project is to analyse how to model surface instabilities correctly to be able to study scale effects related to surface instabilities. During the planned BreaKin CFD JIP, part of the model test results of the BreaKin JIP will be reproduced in CFD to further understand the physical processes responsible for the observations described above. In CFD, parameters can be systematically varied one-by-one to assess their individual influence, and detailed visualisations can provide in-depth understanding of the flow phenomena.

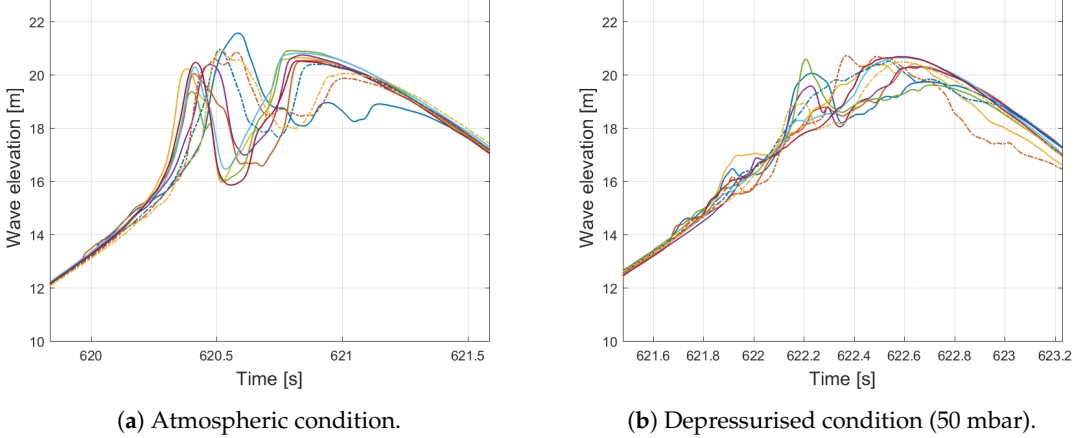

(**a**) Atmospheric condition.

(**b**) Depressurised condition (50 mbar).

**Figure 22.** Repeatability of focused Wave 4 at scale 50 in atmospheric and depressurised condition.

## 5. Conclusions and Future Work

### 5.1. Conclusions

Important steps were made in recent years, to link experimental and numerical wave modelling. In the presented examples, the main focus was on numerical CFD codes, but there is definitely a role for faster second-order, higher-order or fully nonlinear potential flow wave models. Especially in screening studies that require assessment of all wave conditions in a scatter diagram and a large number of wave realisations, such tools can be very valuable. Some conclusions can be drawn from the example studies in the present publications:

1. The kinematics in breaking wave crests can be quite accurately calculated using single-phase CFD without a turbulence model, although the kinematics in the overturning area are somewhat underestimated compared to PIV measurements (which is probably associated with air inclusions).

2. Scale effects and the effect of air pressure in steep and breaking waves are small, as long as there are no air inclusions or surface instabilities.

3. The above conclusions combined with a case study lead to the conclusion that experimental wave impact loads on a floating structure can be accurately reproduced in CFD, if the wave conditions close to the structure are accurately reproduced and the impact is not associated with air inclusions or surface instabilities. This accurate iterative wave event reconstruction is only required for validation or studies that require an exact match with experiments.

4. CFD is too expensive to analyse full scatter diagrams, but quite accurate results within reasonable computational time can be obtained with CFD for 0.5–3 h wave realisations at zero and forward speed.

5. The increasing capability to mutually reproduce wave events from different levels of numerical tools and experiments is promising for the future of screening approaches (where lower-order fast numerical tools are used to identify critical events that can be studied using detailed tools such as CFD and experiments). Still, this requires solid validation that the selected fast screening tool is conservative for the studied problem (it cannot miss critical events), but this is outside the scope of the present study.

6. The increasing capabilities of numerical tools also raise the bar for experimental facilities, as deterministic comparison directly shows for instance when basin reflections arrive at the measurement location. There is a challenge for experimental facilities to reduce unwanted basin effects, as well as for numerical facilities to accurately model basin constraints (to enable validation studies).

There are some challenges common to all wave reproductions, from experiments to different levels of numerical tools and vice versa: how to deal with long-term statistics and durations, absorption and reflection on half-submerged beaches, basin effects (reflections, wave-induced currents, basin modes, spurious and other unwanted waves, shallow-water effects, ...) and wave breaking. Additional studies with the objective to further improve the understanding and link between numerical and experimental wave modelling are planned for the coming years. A few examples are briefly described below.

### 5.2. Future Work: Basin Wave Generation Using CFD Flap Motions

The present 'normal' wave generation procedure assumes linear wave propagation from a target location to the wave generator and applies the Biésel transfer function [14] to calculate the corresponding flap motions. A wave event can be iteratively corrected based on measurements at the target location. Such a procedure was for instance used to reproduce the New Year's Wave in [18]. The wave input can thus be calculated very fast and efficiently for long time traces, but the iterative correction procedure is time-consuming in the basin. The idea of the future work is that the procedure can be improved using CFD or other nonlinear wave models. The wave event will

be modelled iteratively in CFD, using the distance from the inflow to the target location as in the basin. The elevation at the inflow can be used to derive the corresponding flap motions, similar as in the linear procedure. Such an approach requires more time-consuming CFD calculations prior to wave generation, and it can only be used for short wave events (as the basin effects are not included). However, it could reduce the required correction time in the basin, and the wave event will be directly available both in CFD and in the basin.

### 5.3. Future Work: Numerical Shallow-Water Basin Effects

Wave modelling on shallow water is challenging. In shallow water, bound low-frequency wave energy (set-down) increases significantly compared to deep water, while natural frequencies of ship mooring systems may be in the same frequency range [20]. This is also the case at sea at for example near-shore LNG terminals. In most basins, additional shallow-water effects play a role, however, such as the possible generation of low-frequency free waves at a change in basin bathymetry (such as the ramp on the side of a movable basin floor) or resonant basin modes (standing waves) that are in a similar frequency range for the most common basin dimensions [22]. Such basin modes are usually only problematic on shallow water, as their frequency decreases with decreasing water depth (waves travel slower) and the low-frequency excitation increases due to the increased set-down. Active reflection compensation (ARC) algorithms based on instantaneous wave measurements usually have a notch filter around the basin mode frequencies to prevent instability in the system. Attempts at reducing some of the unwanted low-frequency wave content in the basin with deterministic anti-waves were reasonably promising [34–36]. However, the results differed for different modes and wave conditions and such a procedure requires an extensive iterative measurement campaign prior to tests with a model. In wave experiments, many different basin effects occur simultaneously on shallow water (e.g., multiple basin modes simultaneously, possibly in two or more directions, wave reflection on the beaches that are less efficient at damping long waves, free wave generation on the floor ramp, spurious waves from the wave generator). Based on measurements, it is not easy to isolate and study each of the effects separately. Present CFD codes seem mature enough to evaluate wave propagation over a sloping ramp (as demonstrated with CFD code Reef3D [79] in a yet unpublished study), spurious waves generated by the shape of the wave flaps, and maybe even wave absorption and reflection on a sloping beach. This would make it possible to study these effects separately, identify the most important sources of unwanted low-frequency waves and propose ways to mitigate them. A start was made in 2019 to prepare a digital twin of MARIN's Offshore Basin that can be modified to include or exclude certain phenomena in different CFD codes. This work will be continued in 2020, focussing mainly on shallow-water effects.

### 5.4. Future Work: Realistic Waves in Real-Time Simulations

An important application of hydrodynamic research is real-time simulation, used to train new crew or to train crew for specific operations in a safe and controllable environment. For operations with small fast ships on open sea and close to the coast, wave interaction with the bathymetry and the shore and wave steepness are important for operation. This requires real-time simulation of realistic wave fields, including the wave reflection, diffraction, shoaling, refraction and even wave breaking relevant for training. Visualisation of for instance breaking wave crests may also be very important for a successful training. A four-year research project in cooperation with the Dutch Navy was started in 2019, with the objective to improve the wave modelling on MARIN's Fast Small Ship Simulator. This will enable training the crew to operate fast small ships in prescribed environmental conditions in different coastal areas, while at the same time reducing the number of exposed hours on a moving platform of the trainers. This project includes an investigation of the important aspects of such wave fields for the training goals, of the relevant interactions with the ship, different existing and modified wave tools and the options to run them real-time including the ship.

**Author Contributions:** Introduction and abstract, S.v.E. and J.S.; Conceptualisation, methodology, analysis and investigation (CMAI) of basin aspects (Section 2), S.v.E. and J.S.; CMAI of numerical aspects (Section 3), T.B., J.S. and S.v.E.; CMAI of Example 1 (Section 4.1), H.B., S.v.E. and J.H.; CMAI of Example 2 (Section 4.2), T.B., J.H. and B.D.; CMAI of Example 3 (Section 4.3), B.D., R.H. and J.S.; CMAI of Example 4 (Section 4.4), J.S.; Conclusions and future work, S.v.E.; writing—original draft preparation, S.v.E., J.S., T.B., B.D., H.B., R.H. and J.H.; writing—review and editing, S.v.E., J.S., T.B., B.D. and H.B.; project administration, S.v.E.; funding acquisition, S.v.E., J.S., T.B., B.D. and R.H. All authors have read and agreed to the published version of the manuscript.

**Funding:** The breaking wave kinematics work in Section 4.3 was partly funded by the TKI-allowance of the Dutch Ministry of Economic Affairs.

**Acknowledgments:** The authors would like to thank the Cooperative Research Ships (CRS) for the use of the container ship model test data in Sections 4.1 and 4.2.2, the WiFi JIP for the use of the shallow-water model test data in Section 4.2.1 and the BreaKin JIP for the use of the scale effect model test data in Section 4.4.

**Conflicts of Interest:** The authors declare no conflict of interest.

## Abbreviations

The following abbreviations and symbols are used in this manuscript:

| | |
|---|---|
| ARC | Active Reflection Compensation |
| CFD | Computational Fluid Dynamics |
| CFL | Courant (Friedrichs Lewy) number |
| CPU | Central Processing Unit |
| CRS | Cooperative Research Ships |
| DWB | Depressurised Wave Basin |
| JIP | Joint Industry Project |
| KCS | KRISO Container Ship |
| LES | Large Eddy Simulation |
| MARIN | MAritime Research Institute Netherlands |
| OB | Offshore Basin |
| PIV | Particle Image Velocimetry |
| RF | Research Flume |
| RMS | Root-Mean-Square |
| SMB | Seakeeping and Manoeuvring Basin |
| SWB | Shallow Water Basin |
| $\gamma$ | Peak enhancement factor [-] |
| $H_s$ | Significant wave height [m] |
| $t$ | Time stamp [s] |
| $T_p$ | Peak wave period [s] |
| $u$ | Horizontal velocity [m/s] |
| $x$ | Horizontal $x$-coordinate [m] |
| $z$ | Vertical $z$-coordinate [m] |

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
