# Peer review of "Linking Experimental and Numerical Wave Modelling"

_jmse, doi:10.3390/jmse8030198_

Round 1

Reviewer 1 Report

The manuscript presents numerical modelling using ComFLOW and CFD- ReFresco and comparison to model test data. The objective is to demonstrate the possibilities and challenges in accurately reproducing the results from model tests in numerical models of different fidelities, with focus on CFD modelling.

The manuscript presents the topic well and the discussion meets the objectives set for the paper. The consideration of basin effects in the numerical model is also discussed. 

Some explanation for the mismatch at the peak frequency seen in Fig 7 should be given. This does not seem to be due to grid resolution as higher frequencies are well represented. The spectrum matches the measured spectrum for all other frequencies as well. This would then imply a lower total energy in the numerical model. Could measurements be a source of this mismatch? The authors need to comment on this.

Time stamps are necessary for Fig 17.

Cosmetic: Fig 17b seems to encroach on the caption for Fig 17a.

The manuscript is well written and the reviewer recommends acceptance with these minor revisions

Author Response

Dear reviewer,

Thank you very much for your effort and useful comments. Our replies are assembled in the attached document.

Kind regards,

Sanne van Essen

Reviewer 2 Report

General comment:

The paper deals presents the recent advantages of MARIN regarding experimental and numerical wave modelling, with special emphasis on hybrid modelling. The paper includes different examples on this topic.  

As a general comment, the paper is well written and it contains interesting examples, but it needs some improvements to be suitable for publication.

First of all, it is difficult to know what is new in this manuscript compared to the many works of the authors included as references. I think it is necessary to make this point clear in the manuscript so that it is considered a research article and not a review article. In addition, it is surprising that more than 70% of the references are self-citations of the authors of the manuscript, and more than 95% are works included in conference proceedings or technical reports (only one article in a journal in the references list). I think that works by other authors as well as articles included in scientific journals should be included in the state of the art.

On the other hand, it is quite confusing the way of organizing the information with 4 different sections regarding some examples. I suggest all the examples must be included in the same section as subsections. There are also numerous cross-references between sections, which makes reading the manuscript confusing.

In general the quality of the figures must be improved.

Specific comments:

  1. Page 6 and 7, figure 4 and 6: it's difficult to identify each case in the colored lines. I suggest including marker or line types that make it easier to identify any case within the figure.
  2. Page 8, figure 7: please improve the quality of right panel figure.
  3. Page 9, figures 8 and 9: please improve quality.
  4. Page 10, figure 11: please consider increasing the size of the legends and axes labels.
  5. Page 13, figures 15 and 16: please improve quality.
  1. Page 14, fig 17: the lower panel is covering the legend on the upper panel.

Author Response

Dear reviewer,

Thank you for your effort and useful comments. Our replies to your comments can be found in the attached document, and changes can be found in the updated draft of our paper.

Kind regards,

Sanne van Essen

Round 2

Reviewer 2 Report

I believe that all the changes and modifications suggested have been made and that the article now reads more clearly. The quality of the images has also been greatly improved. Congratulations for the effort made.